

# A systematic evaluation of 15 actual evapotranspiration formulations within conceptual hydrological models

Gabrielle Burns[1], Keirnan Fowler[1], Murray Peel[1], Clare Stephens[2]

[1]Department of Infrastructure Engineering, The University of Melbourne, Parkville, VIC Australia

[2]Hawkesbury Institute for the Environment, Western Sydney University, Richmond, NSW, Australia

*Correspondence to*: Gabrielle Burns (gabrielle.burns@unimelb.edu.au)

**Abstract.** Actual evapotranspiration (AET) is a major component of the water balance, yet it is rarely assessed for accuracy in conceptual rainfall-runoff models that are often calibrated to match streamflow only. Inaccurate representation of underlying AET processes may cause models to incorrectly simulate long-term changes in partitioniapendng between AET and
streamflow, even if this partitioning was relatively accurate during calibration. To investigate AET representation within conceptual hydrological models, we systematically tested 15 evapotranspiration (ET) equations that convert potential evapotranspiration (PET) and soil moisture to AET. The 15 equations represent common practice, having been sourced from a published comprehensive review of conceptual hydrological models. Each of these 15 formulations were trialled within three conceptual hydrological models (GR4J, Simhyd and Vic). Following multi-objective calibration, we evaluated performance
across both streamflow and flux tower AET measurements at seven catchments from a range of Australian climates. A small number of AET equations outperformed the rest, with one equation standing out, which uses a non-linear relationship with soil moisture storage and can scale down AET such that it cannot equal PET. This equation achieved a higher objective function value for both AET and streamflow and accurately captured evapotranspiration signatures. However, even this equation showed limitations in reproducing observed AET, suggesting persistent issues across commonly used formulations. These
shortcomings may reflect missing vegetation-related dynamics and other simplifications. Our findings highlight the importance of ET equation selection in modelling AET and streamflow, and we recommend the identified equation as a promising option for future Australian studies. Further work is needed to test equations for consistency with known processes to improve the physical realism of conceptual hydrological models.

## 1 Introduction

Hydrological models play a critical role in understanding and replicating catchment behaviours, providing insights into the movement and storage of water within landscapes (Liu et al., 2017). These models are widely applied in water resource management, where they inform decision-making in contexts such as flood mitigation, drought planning, and adapting to the impacts of a shifting climate (Grigg & Hughes, 2018). Streamflow is often the main output of interest, however, internal fluxes such as soil moisture, groundwater recharge, and actual evapotranspiration (AET) are also simulated. Hydrological models are



commonly calibrated to match observed streamflow data alone, perhaps based on the assumption that streamflow is the culmination of the other catchment processes and accurate streamflow replication implies accuracy across all modelled fluxes. In reality, models may exhibit poor replication of internal fluxes like actual evapotranspiration (AET) (Kelleher & Shaw, 2018) or streamflow generation mechanisms, despite achieving acceptable streamflow, due to equifinality - where multiple parameterisations or process representations yield similar outputs (Beven, 2006; Khatami et al., 2019). The wide range of

internal behaviours shown in flux maps by Khatami et al. (2019) illustrates this issue and raises concerns about how reliably models capture underlying hydrological processes.

Increasing emphasis has been placed on the ability of models to operate under change, particularly climate change. Blöschl et al. (2019) highlighted the need for hydrological models capable of adapting to changing conditions, including evolving vegetation dynamics. For example, inaccurate representation of underlying processes may cause models to incorrectly simulate

long-term changes in water partitioning, despite being accurate during calibration. Furthermore, studies have found that model performance decreases when multi-annual shifts in rainfall-runoff relationships occur, often driven by long-term changes in climate forcing (Saft et al., 2016). These historical cases provide valuable analogues for understanding how models may perform under future climate change. Vegetation may play a key role in driving this non-stationary catchment behaviour, yet it is rarely incorporated into traditional conceptual model structures (Deb & Kiem, 2020). Furthermore, vegetation dynamics,

critical to AET processes, are rarely considered explicitly, creating significant gaps in understanding and model performance (Duethmann et al., 2020). Neglecting the role of vegetation in these processes further exacerbates the limitations of existing models. The hydrology community has identified these challenges as critical to advancing the field, suggesting current models need improving (Fowler et al., 2020; Stephens et al., 2019; Vaze et al., 2010). Addressing these gaps requires a shift toward harmonising conceptual models with process-based understanding to improve model adaptability under change. However, a

useful first step is evaluating the performance of existing empirical equations used within models to determine which best capture key processes.

Conceptual hydrological models typically estimate AET using potential evapotranspiration (PET) as an upper limit, with reductions based on water availability and model-specific assumptions. The specific choice of equations can influence how accurately AET is represented, but explicit evaluation of AET accuracy remains uncommon (Kelleher & Shaw, 2018), and

few studies stray from the default AET equations of their chosen model. In recent years, studies have increasingly adopted multi-objective calibration approaches, some incorporating AET alongside streamflow to enhance model performance. Research demonstrates that integrating AET into calibration not only enhances the accuracy of AET estimates (Arciniega-Esparza et al., 2022; Dembélé et al., 2020; Herman et al., 2018; Rientjes et al., 2013) but can also improve the simulation of total water storage (Bai et al., 2018; Pool et al., 2024). Despite these advances, the characterisation of AET accuracy may be

limited to the consideration of aggregate measures of performance, which led Gardiya Weligamage et al. (2025) to propose the use of "signatures", a concept borrowed from streamflow evaluation (e.g. McMillan, 2021), to separately characterise different aspects of AET dynamics.



Few studies have systematically compared AET equation options in a controlled way. For example, previous research has
investigated which AET products are best to use in calibration (Taia et al., 2023) and assessed the impact of using different
PET forcing inputs (Bai et al., 2016), but the impact of the equations themselves has not been investigated. Studies examining
the performance of modelled AET are often confined to assessing the default equations built into specific hydrological models
(Arciniega-Esparza et al., 2022; Dembélé et al., 2020; Guo et al., 2017; Herman et al., 2018; Rientjes et al., 2013). This
approach makes it difficult to disentangle whether differences in performance are from the AET equation itself or from the
surrounding model structure. By systematically evaluating AET equations in isolation, independent of broader model-specific
assumptions, we can better understand their individual strengths and limitations. Additionally, many commonly used equations
may already encode elements of process understanding that are not immediately apparent. Assessing their empirical
performance in a controlled framework could reveal implicit process representations, offering insights that inform both
conceptual and process-based model development. This is particularly important given the continued reliance on conceptual
models in many water resource applications.

The aim of this study is to evaluate the performance of different AET equations in conceptual hydrological models, focusing
on their ability to reproduce observed AET, while also considering the impact (if any) on streamflow simulation. This study's
novelty lies in its systematic evaluation of different AET equations while holding other aspects of the model structure constant.
In addition, novel aspects include the wide range of different AET equations tested, the use of multi-objective calibration
incorporating flux tower derived AET data, and the use of AET signatures in model evaluation.

The AET equations are based on the study of Knoben et al. (2019) who conducted a systematic review of 47 existing conceptual
rainfall runoff models, compiling them into the Modular Assessment of Rainfall-Runoff Models Toolbox (MARRMoT)
framework. This framework's structure isolates the unique equations used for a particular process such as AET—i.e. although
there are 47 models, there are not 47 different AET equations because multiple models may use the same AET equation. This
unique MARRMoT list forms the basis of the AET equations tested here. In addition, MARRMoT is used for the simulation
experiment itself. It provides an effective framework for addressing these challenges because it is set up to ensure consistent
model implementation, allowing for intercomparison and modification of model structures and internal components. By
leveraging MARRMoT, the performance of various AET equations can be systematically investigated within consistent
conceptual model structures, addressing the gaps identified above. By advancing the representation of AET, this study
contributes to more robust and reliable hydrological modelling practices particularly under changing environmental conditions,
where traditional model assumptions are increasingly challenged.

## 2 Methods

### 2.1 Overview of methodology

As noted above, a key limitation in previous studies is that comparisons of AET performance have been made between entire
models, rather than isolating the contribution of individual AET equations. Thus, it is impossible to tell whether differences





in performance are due to the AET equation used within the model or the surrounding model structure. To address this, we systematically substitute a range of AET equations into a fixed model structure, testing each in turn. To ensure the findings are not model-specific, this process is repeated across three commonly used conceptual hydrological models, treating each model as a consistent "container" for testing the equations.

Specifically, (as indicated in Fig. 1), this study systematically evaluates 15 evapotranspiration (AET) equations by substituting them into the models, applied across seven catchments. Each model is individually calibrated ($15 \times 3 \times 7 = 315$ calibrations) using a multi-objective function that equally weights the match to both streamflow and flux tower-derived AET data.

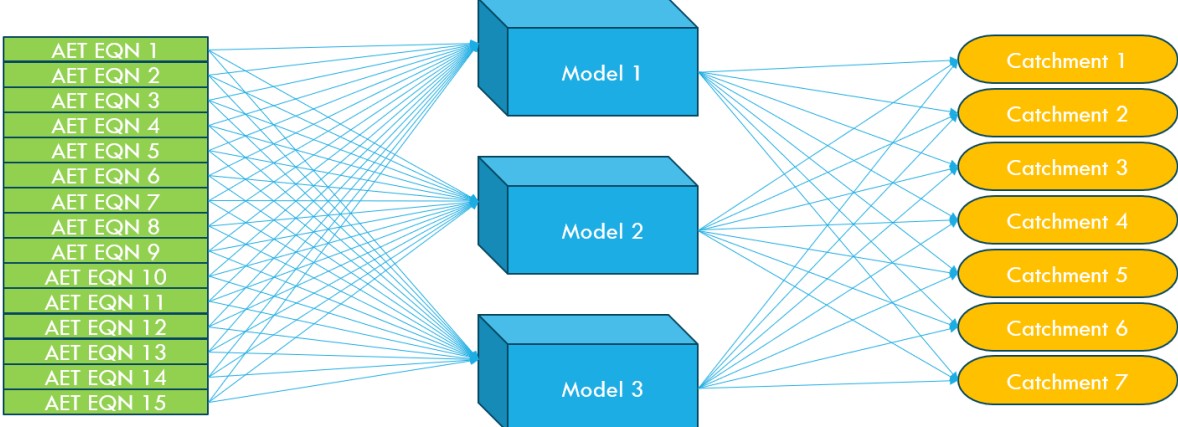

**Figure 1: General methodology visualisation. By holding the model structure constant and only varying the AET equation, this**
**approach isolates the influence of different AET equations on model performance.**

We utilise flux tower data as our benchmark for evaluating actual evapotranspiration (AET) in hydrological models because it provides direct, high-temporal resolution observations of surface energy and water fluxes (Beringer et al., 2022). Compared to remotely sensed AET estimates, which offer broader spatial coverage, flux tower data avoids uncertainties related to satellite retrieval algorithms and coarse spatial and temporal resolution, including uncertainty around overpass timing, which may not
align well with daily model time steps. Additionally, Gardiya Weligamage et al. (2025) assessed the quality of remotely sensed AET across Australia and identified several limitations of remotely sensed AET, including seasonal inconsistencies and variability issues. The flux tower data used here is sourced from the TERN-OzFlux (Terrestrial Ecosystem Research Network) dataset, which is the Australia and New Zealand portion of the Fluxnet network of stations that measure carbon, water, and energy fluxes (Beringer et al., 2022).
Nonetheless, flux tower measurements are point-based, while hydrological models simulate catchment-averaged fluxes, leading to a scale mismatch that requires careful consideration. A recent study by Gardiya Weligamage et al., (2025) paired 15 OzFlux sites with nearby catchments from the CAMELS-AUS dataset (Fowler et al., 2024) and evaluated simulations from several models. To further address this issue, we applied strict selection criteria, focusing only on flux tower–catchment pairs where we had reasonable confidence in representativeness. In addition to geographic proximity, we required close alignment





in precipitation and temperature regimes, and placed particular emphasis on land cover similarity, given the importance of vegetation dynamics in controlling AET. Sites were excluded where vegetation type, structure, or density diverged markedly between the tower footprint and the broader catchment. This process aimed to reduce discrepancies between observed and modelled AET that could otherwise result from mismatched biophysical controls.

While we acknowledge the scale limitations of point-based flux data, supplementary analysis from Gardiya Weligamage et al.

(2025, in prep) suggests that even distant flux tower observations (up to hundreds of kilometres away) often provide more realistic AET estimates than co-located remotely sensed data. This highlights the value of flux tower measurements for hydrological model evaluation, particularly when accompanied by rigorous site selection.

## 2.2 Study area

The selected flux tower sites represent a range of climatic zones and vegetation types characteristic of the diverse Australian

hydrological conditions, as outlined in Table 1, and shown in Fig. 2. Their selection ensures that the findings are applicable to broader Australian conditions.

**Table 1: Details on catchments and flux tower sites**

| Flux tower site name | Flux tower site description | Flux tower site loc. | Nearby catchm-ent code | Catchm-ent size (km²) | Mean daily P (mm/d) | Mean daily PET (mm/d) | Australian State or territory | Season high P / low P * |
|---|---|---|---|---|---|---|---|---|
| **Wombat Forest** | Mixed Eucalyptus regrowth forest | -37.423, 144.094 | 407221 | 167.5 | 2.190 | 2.951 | Victoria | JJA / DJF |
| **Whroo** | Box woodland forest | -36.673, 145.029 | 405229 | 108.7 | 1.441 | 3.317 | Victoria | JJA / DJF |
| **Tumbar-umba** | Wet sclerophyll, alpine ash forest | -35.657, 148.152 | 401009 | 215.5 | 2.822 | 3.215 | New South Wales | JJA / DJF |
| **Robson Creek** | Tropical rainforest | -17.117, 145.630 | 111007A | 521.4 | 7.110 | 4.702 | Queens-land | DJF / SON |
| **Litchfield** | High rainfall, frequently burnt tropical savanna | -13.179, 130.795 | G8150180 | 1044.2 | 3.901 | 5.540 | Northern Territory | DJF / JJA |
| **Dry River** | Open forest savanna | -15.259, 132.371 | G8140011 | 4794.5 | 2.065 | 5.342 | Northern Territory | DJF / JJA |
| **Gingin** | Coastal heath Banksia woodland | -31.376, 115.714 | 617003 | 1404.1 | 1.807 | 4.067 | Western Australia | JJA / DJF |

* Southern hemisphere seasons: DJF ("summer"), MAM ("autumn"), JJA ("winter"), SON ("spring")



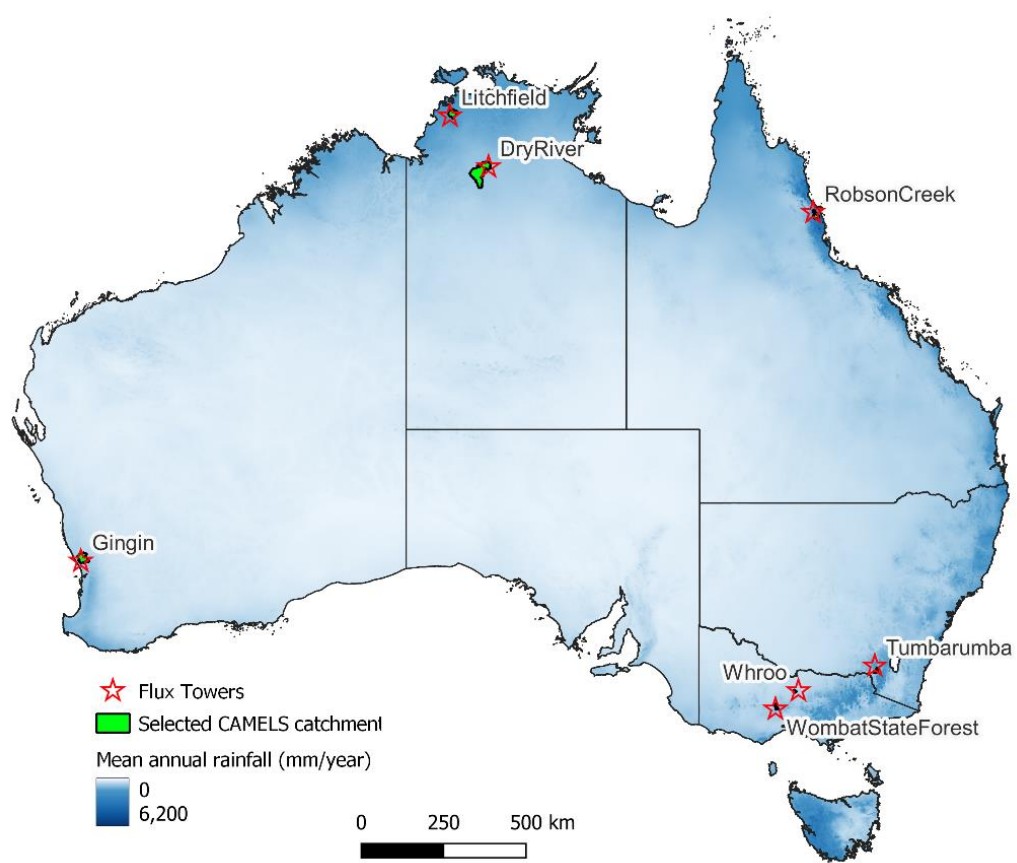

**Figure 2: Map of Australia showing the seven flux tower sites and their associated catchments.**

### 2.3 Data

OzFlux eddy covariance evapotranspiration data were accessed from OzFlux towers through the Terrestrial Ecosystem Research Network (TERN) data portal (https://portal.tern.org.au/, 2024 Version 2, last accessed 07/03/2025) using level 6, quality controlled and gap filled daily time scales.

Remaining hydrometeorological data were sourced from the CAMELS-AUS dataset, as follows. Potential evapotranspiration data (specifically Morton's Wet Environment Evaporation) are provided in CAMELS and sourced from the Scientific Information for Land Owners, or SILO, database, published by the state of Queensland (https://www.longpaddock.qld.gov.au/silo/, last accessed 15/01/2025). Precipitation data were provided in CAMELS and sourced from the Australian Gridded Climate Data (AGCD) dataset of the Bureau of Meteorology (https://portal.ga.gov.au/, last accessed 15/01/2025).



## 2.4 Flux tower data adjustment

Building on the careful selection of representative tower-catchment pairs (Section 2.1), we further adjusted the flux tower AET data to improve comparability with catchment-scale values. Since actual evapotranspiration at the catchment scale is not directly measurable, we applied a long-term water balance approach to provide an independent estimate of long-term catchment-average AET and adjust the flux tower data accordingly. This adjustment involved calculating a linear scaling factor that was applied to the full flux tower AET time series, correcting the long term-average without altering the intra-annual shape of the AET curve. This approach assumes that while the temporal dynamics of AET at the flux tower (e.g. seasonal dynamics) are broadly representative of catchment-scale behaviour, the magnitude may differ due to spatial heterogeneity in vegetation, soil properties, or microclimate.

To derive the scaling factor via catchment water balance, we assumed negligible long-term change in storage ($\Delta S \approx 0$). This is a common assumption in hydrology (e.g. Weligamage et al., 2024), justified here by the fact that the water balance components: precipitation (P), streamflow (Q), and AET, are aggregated over a multi-year period. As such, any difference in storage between the start and end of the period is small relative to the cumulative fluxes and has minimal influence on the resulting estimate. The catchment water balance therefore simplifies to:

$$\Delta S = P - Q - AET \quad \rightarrow \quad AET_{catchment} = P - Q, (1)$$

where P is long-term average precipitation and Q is long-term average streamflow, both measured over the same period as the flux tower record. The corresponding long-term average AET from the flux tower, $AET_{tower}$, was calculated from the eddy covariance measurements. A scaling factor f was then derived as:

$$f = \frac{AET_{catchment}}{AET_{tower}}, (2)$$

This factor was applied to the full flux tower time series:

$$AET_{adjusted}(t) = f \times AET_{tower}(t), (3)$$

This adjustment ensures that the flux tower data matches the long-term magnitude of AET estimated at the catchment scale, while preserving its temporal variability. Combined with careful catchment selection, this was deemed the most practical method available to reduce scale mismatch and increase confidence in data used for the model evaluation.

## 2.5 Models and equations

This experimental setup, introduced in Section 2.1, was applied across three model structures - GR4J, Simhyd, and VIC - which were selected from the 47 options in MARRMoT to represent a range of complexity and conceptual assumptions. GR4J is the most parsimonious model, with four parameters and two representative storages (Perrin et al., 2003). Simhyd represents an intermediate level of complexity, with seven parameters and three storages (Chiew et al., 2002). VIC is the most complex model chosen, comprising ten parameters and three storages (Liang et al., 1994). In Simhyd and VIC, AET is drawn from multiple storages, and the evaluation of AET simulations was always conducted on the sum of AET across these storages. However, this still leaves the question of which storage is subject to the experimental changes in equations. For the purposes



of this study, the AET equation was substituted into the storage that, based on prior understanding of each model's design, contributes most substantially to total AET. The remaining storages retained their original AET formulations. This decision

was made a priori, recognising that the contribution of each storage may vary depending on parameter values. As stated, calibration and evaluation performance scores are based on the total modelled AET, regardless of the internal distribution across storages.

Across the 47 models in MARRMoT there are 23 unique AET equations, but the experiment did not test all 23 because some are incompatible with the three chosen models, and also because we found some redundancy between the equations. For

instance, some equations used different names for equivalent parameters, such as maximum storage represented as "Smax" in one equation and "S3 parameter" in another. When simplified, these equations were found to have the same functionality. Additionally, two equations were excluded because they were designed to operate on soil moisture storages defined in a "deficit" manner, which was not compatible with the three selected models. Following the process of simplifying equations to ensure compatibility with the chosen models and removing redundant parameters, 15 of the 23 equations remained. These

equations retained their original naming according to the MARRMoT framework.

Table 2 provides an overview of the selected 15 equations, including their corresponding number within MARRMoT, the number of additional parameters required, descriptions as provided in MARRMoT, and simplified formulas. Additionally, Table 3 presents the list of models that utilise each equation. The resulting 15 AET equations represent diverse approaches to modelling the conversion of potential evapotranspiration (PET) into actual evapotranspiration (AET) as a function of soil

moisture. This diversity of equations allows for a robust experiment that represents the diversity of current practice in conceptual modelling. These approaches fall into five major relationship types, which are summarised below:

1. Linear relationships with soil moisture (equations 1, 7 and 11) assume a direct proportionality between soil moisture and evapotranspiration.

2. Threshold-based relationships (6, 8, 16, 21, 22, and 23) impose thresholds (e.g., wilting points or storage thresholds)
that control when and how PET translates into AET.

3. Nonlinear relationships with soil moisture (4, 13, 19, and 20) include non-linear scaling factors to represent more complex vegetation or soil interactions.

4. Multi-component representations (3, 20, 23) explicitly separate processes such as transpiration from vegetation and evaporation from bare soil.

5. Evaporation-rate limitations (20 and 22) cap AET to a maximum rate or constrain it below certain thresholds

**Table 2. Description of the 15 AET equations. P1 and p2 are additional parameters, whereby [0-1] indicates the parameter is set between 0-1, and [mm] indicates it is an unbound parameter with the value representing millimetres. \*\*\*The SmoothStorageThresholdFunction(S, 0.01) refers to a logistic smoothing function implemented in MARRMoT, which gradually reduces fluxes as storage (S) approaches a lower threshold. This avoids abrupt cut-offs by applying a smooth, continuous transition,**
**governed by a steepness parameter (here, 0.01), allowing for more numerically stable and physically plausible flux behaviour when storage is low.**



| Equation no. (from MARR MoT) | Extra params | Description | Formula simplification | Features | | | | |
|---|---|---|---|---|---|---|---|---|
| | | | | Linear scaling | Quadratic scaling | Exponential scaling | Nonlinear scaling | Threshold based |
| 7 | 0 | Evaporation scaled by relative storage | Minimum of <br> • S / Smax * PET <br> • S | X | | | | |
| 11 | 0 | Evaporation quadratically related to current soil moisture | $(2 \times (S / Smax) - (S / Smax^2)) \times PET$ | | X | | | |
| 1 | 0 | Evaporation at the potential rate | Minimum of <br> • S <br> • PET | X | | | | |
| 16 | 1 | Scaled evaporation if another store is below a threshold | Minimum of <br> • p1[0-1] ×PET × SmoothStorageThresholdFunction (S, 0.01) *** <br> • S | | | | | X |
| 2 | 1 | Evaporation at a scaled, plant-controlled rate | Minimum of <br> • p1[mm] × S / Smax <br> • PET <br> • S | X | | | | |
| 19 | 2 | Non-linear scaled evaporation | Minimum of <br> • S <br> • PET <br> • p1[0-1] × PET × S / Smax$^{p2[0-1]}$ | | | | X | |
| 21 | 2 | Threshold-based evaporation with constant minimum rate | Minimum of <br> • Maximum of <br>   o p1[mm] × p2[0-1] <br>   o PET × minimum of <br>     ▪ S / p1[mm] <br>     ▪ 1 <br> • S | | | | | X |
| 23 | 2 | Transpiration from vegetation at the potential rate if storage is above field capacity and scaled by relative storage if not, addition of evaporation from bare soil scaled by relative storage | Minimum of <br> • PET / Smax × PET <br> • PET × S / (p1[0-1] × Smax) + S / Smax × PET <br> • S | X | | | | X |
| 4 | 2 | Constrained, scaled evaporation if storage is above a wilting point | Minimum of <br> • PET × p1[0-1] × (S – p2[0-1] × Smax) / (Smax – p2[0-1] × Smax) <br> • S <br> • 0 | | | | X | X |





| 3 | 2 | Evaporation based on scaled current water storage and wilting point | Minimum of<br>• $S / (p1[0-1] \times Smax \times PET)$<br>• PET<br>• S | X | | | | |
| 6 | 2 | Transpiration from vegetation at the potential rate if storage is above a wilting point and scaled by relative storage if not | Minimum of<br>• $PET \times S / (p1[0-1] \times Smax)$<br>• PET<br>• S | X | | | | |
| 8 | 2 | Transpiration from vegetation, at potential rate if soil moisture is above the wilting point, and linearly decreasing if not. Also scaled by relative storage across all stores | Minimum of<br>• Minimum of<br>   ○ PET<br>   ○ $S / p1 [mm] \times PET$<br>   ○ S<br>• 0 | X | | | | X |
| 13 | 2 | Exponentially scaled evaporation | Minimum of<br>• $p1[0-1]^{p2[0-1]} \times PET$<br>• S | | | X | X | |
| 20 | 2 | Evaporation limited by a maximum evaporation rate and scaled below a wilting point | Minimum of<br>• $p1 [mm] \times S \div (p2[0-1] \times Smax)$<br>• PET<br>• S | X | | | | X |
| 22 | 2 | Threshold-based evaporation rate | Minimum of<br>• S<br>• $(S \times p1[mm]) \div (p2[mm] - p1[mm]) \times PET$<br>• PET | X | | | | X |

**Table 3. Models using each AET equation. Note that some models appear under multiple equations due to different equations being used within the model for evaporating from different stores.**

| Equation no. (from MARRMoT) | Models featuring equation |
|---|---|
| 1 | Wetland model, Alpine model v1/v2, Hillslope model, New Zealand model v2, Penman, SimHyd, Large-scale catchment water and salt balance model element, Thames Catchment Model, Flex-I, Tank model, Sacramento-SMA, Flex-IS, MODHYDROLOG, Tank model – SMA, Midland Catchment Runoff Model, NAM, HYCYMODEL, ECHO, PRMS, CLASSIC, Forellenbach model |
| 2 | SimHyd, MODHYDROLOG |
| 3 | Flex-I, Flex-IS, HYCYMODEL, Collie River v2/v3, TOPMODEL, Flex-B, HBV-96 |
| 4 | Plateau model |
| 6 | New Zealand model v1, New Zealand model v2, Susannah Brook v1 |



| 7 | New Zealand model v2, Sacramento-SMA, NAM, PRMS, Collie River v2/v3, United States model v1, Susannah Brook v1, Collie River 1, Susannah Brook model v2, Australia model, VIC, HyMOD, MOPEX-1/-2/-3/-4/-5 |
|---|---|
| 8 | United States model v1 |
| 11 | GR4J |
| 13 | SMAR |
| 16 | Penman, Thames Catchment Model |
| 19 | Large-scale catchment water and salt balance model element (LASCAM), GSM-SOCONT |
| 20 | GSFB |
| 21 | Xinanjiang |
| 22 | ECHO |
| 23 | Forellenbach model (IHM) |

## 2.6 Calibration

The calibration approach was based on single objective optimisation to a composite objective function that equally weighted the performance of the model against streamflow data from CAMELS-AUS and AET data from the flux towers. For streamflow (Q), a bias-penalised objective function was used that has separate components for high flow and low flow performance, as described in Trotter et al. (2022) (Eqn. 4). For AET, the Kling-Gupta Efficiency (KGE) with square root transformation was applied, following Gardiya Weligamage et al. (2025) (Eqn. 5). This calibration approach ensured that both monthly and seasonal dynamics of AET were incorporated, alongside a commonly used method for streamflow. The equations are shown in the list below, and the final objective function value (OFV) was determined by averaging the two objective function values (Eqn. 6).

$$OFV_{Streamflow} = of\_bias\_penalised\_log\,(Q_{observed}\,, Q_{simulated}), (4)$$

$$OFV_{AET} = of\_KGE\_sqrt\,(AET_{observed},\ AET_{simulated}), (5)$$

$$OFV = 0.5 * (OFV_{Streamflow} + OFV_{AET}), (6)$$

Model calibration was performed using the Covariance Matrix Adaptation Evolution Strategy (CMA-ES) algorithm, a high performing algorithm commonly used in hydrology (Arsenault et al., 2014), as implemented in the default calibration framework of MARRMoT v2.1 (Trotter et al., 2022).

## 2.7 Process and data analysis

In order to systematically test the available AET equations, the 15 AET equations were individually substituted into each model. This was repeated for all seven catchments, resulting in 315 calibrations. AET equations were ranked to determine which performed best according to the objective function values described above.





In addition to traditional objective function values, we conduct a more comprehensive assessment of model performance by analysing results based on evapotranspiration signatures, as defined by Gardiya Weligamage et al. (2025). These signatures, similar to commonly used streamflow signatures (e.g. (McMillan, 2021)), provide insights into whether the models can reproduce specific aspects of AET. The eight calculated AET signatures are: long-term median and inter-annual variability for annual dynamics; peak timing and lag-12 correlation for seasonal dynamics; water stress, variability, and synchronicity for

monthly dynamics; and rainfall event responsiveness for daily dynamics.

Finally, we conducted split-sample testing to evaluate how well the updated models responded to previously unseen data. This approach ensured that the identified improvements in model performance were not an artifact of overfitting but instead reflected a more generalisable enhancement in AET representation. For brevity, this test was conducted only on the highest performing AET equation in calibration. The available AET data were divided in half, with calibration performed on the first

half and evaluation on the second and repeated in reverse for a second calibration. This procedure was applied to all catchments and all three models, comparing the original model formulation with the version incorporating the best-performing evapotranspiration equation.

## 3 Results

### 3.1 Overview of results

Figure 3, provides an example of AET model outputs by showing a time series of simulated AET at one of the seven sites, Wombat Forest, using the Simhyd model. In this figure, Simhyd was run with each of the 15 evapotranspiration (AET) equations consecutively, and all other model components were held constant. The observed flux tower AET is shown as a thick blue line, while the model outputs from the various AET equations are displayed in different colours. PET (grey) and rainfall (inverted, light blue) are also shown to illustrate the key drivers of AET and help contextualise model behaviour.

This figure highlights the variation in simulated AET that arises solely from substituting the AET equation, demonstrating sensitive model outputs to this component of the model. Similar outcomes were observed across other catchments and model structures (21 figures total), with the remaining plots included in the supplementary materials (S1) for reference.



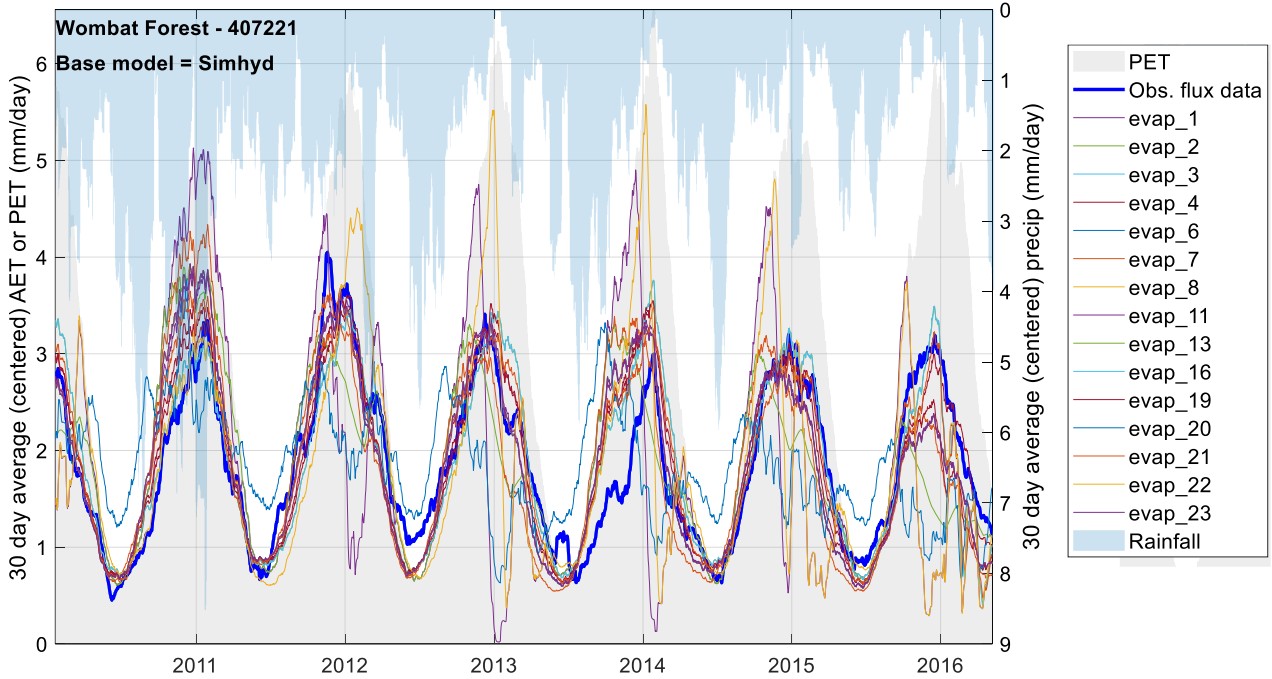

**Figure 3. Observed flux tower data, precipitation, and PET for Wombat Forest, Victoria, Australia. 15 calibrations showing**
**simulated AET at the site (evap_1, evap_2, etc.) for the base model (Simhyd).**

As noted, each AET equation is optimised individually across streamflow and actual evapotranspiration (AET) objective function values (OFV), resulting in 315 calibrations (3 models × 7 catchments × 15 equations). Table 4 displays the summary of these results by ranking the combined objective function values (OFVs) to two decimal places for each model and catchment, with a rank of 1 indicating the best-performing equation. Equal ranks were assigned when the rounded values (to two decimal

places) were equal. Full tables with raw OFV scores and rankings are included in the supplementary materials (S2). The summary shown here aggregates rankings across all catchments for each model and also provides an overall ranking across all models.

Equation 19 emerges as the overall top performer, achieving the highest average rank. Equations 3, 8, and 21 also rank highly, though their performance varies more noticeably across model structures (e.g., Simhyd vs GR4J).

**Table 4. Summary of AET equation rankings (1 = best; 15 = worst) based on combined streamflow and actual evapotranspiration (AET) objective function values (OFVs). Rankings are averaged across all catchments for each model (GR4J, Simhyd, and VIC). The final column shows the overall ranking across all models. An asterisk is placed in each model's column next to its "native" equation's performance. See A2 for more details.**

| Equation | GR4J | Simhyd | VIC | All models |
|---|---|---|---|---|
| 1 | 15 | 14 | 15 | 15 |
| 2 | 12 | 5* | 12 | 12 |
| 3 | 5 | 3 | 8 | 4 |



| | | | | |
|---|---|---|---|---|
| 4 | 11 | 5 | 3 | 5 |
| 6 | 5 | 8 | 11 | 7 |
| 7 | 10 | 12 | 2* | 9 |
| 8 | 5 | 2 | 8 | 3 |
| 11 | 9* | 11 | 5 | 11 |
| 13 | 8 | 10 | 6 | 9 |
| 16 | 7 | 9 | 9 | 11 |
| **19** | **1** | **1** | **1** | **1** |
| 20 | 13 | 13 | 13 | 13 |
| 21 | 2 | 6 | 4 | 2 |
| 22 | 14 | 15 | 14 | 14 |
| 23 | 5 | 8 | 11 | 7 |

To further explore the calibration results, Fig. 4 separates the combined objective function values into their two components:
streamflow and AET performance. Each subplot represents a catchment, showing variations across all models and AET equations (15 points per model, 45 total). The most effective AET equations appear toward the top-right quadrant, indicating strong performance on both streamflow and AET objectives. The native (original) AET equation used in each model is marked with a black star, and equation 19 is highlighted in red for reference. These plots visually represent that while some models exhibit catchment-specific variations (e.g., GR4J's poor performance for Whroo), it is noted equation 19 consistently ranks
among the best.



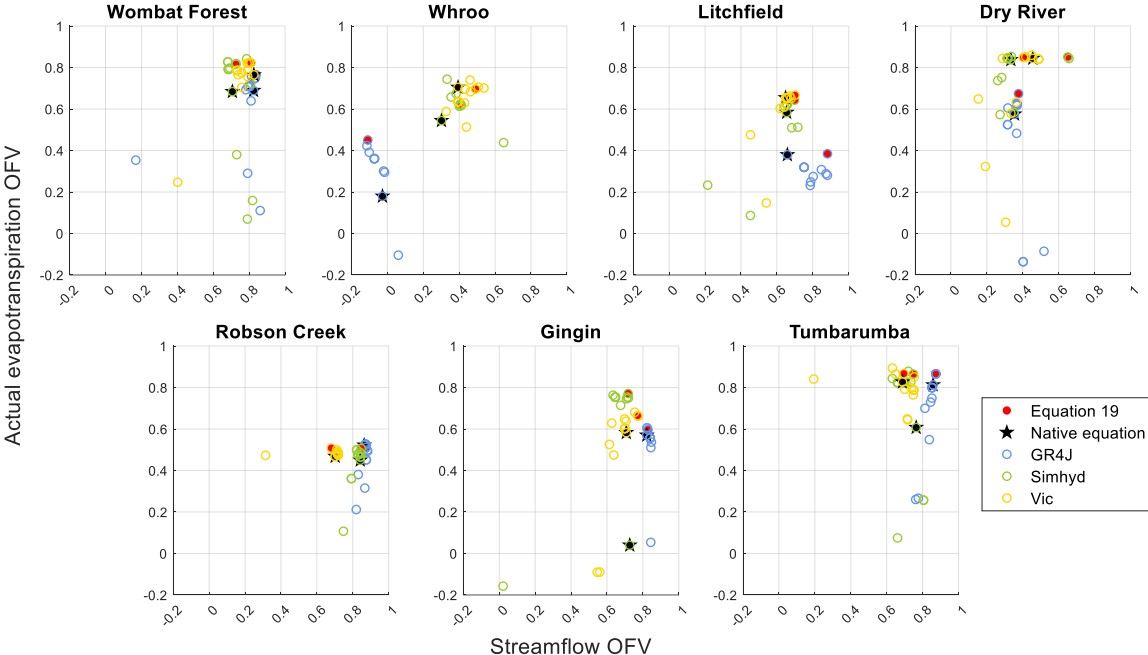

**Figure 4. Trade-off plots of objective function values (OFVs) for streamflow (x-axis) and AET (y-axis), shown for each catchment. Each subplot displays all 45 calibrations (15 AET equations × 3 models), with equation 19 highlighted in red and native AET equations marked with black stars. Higher values indicate better model performance.**

Despite some promising results, certain hydrological behaviours remain poorly represented. For instance, even the four best-performing equations (3, 8, 19, and 21) fail to perform adequately at specific sites. This is evident when comparing results at Tumbarumba and Litchfield (Fig. 5a and 5b, respectively). All models provide reasonable estimates of AET at Tumbarumba, and both Simhyd and VIC perform reasonably well at Litchfield. However, GR4J significantly overpredicts AET at Litchfield.



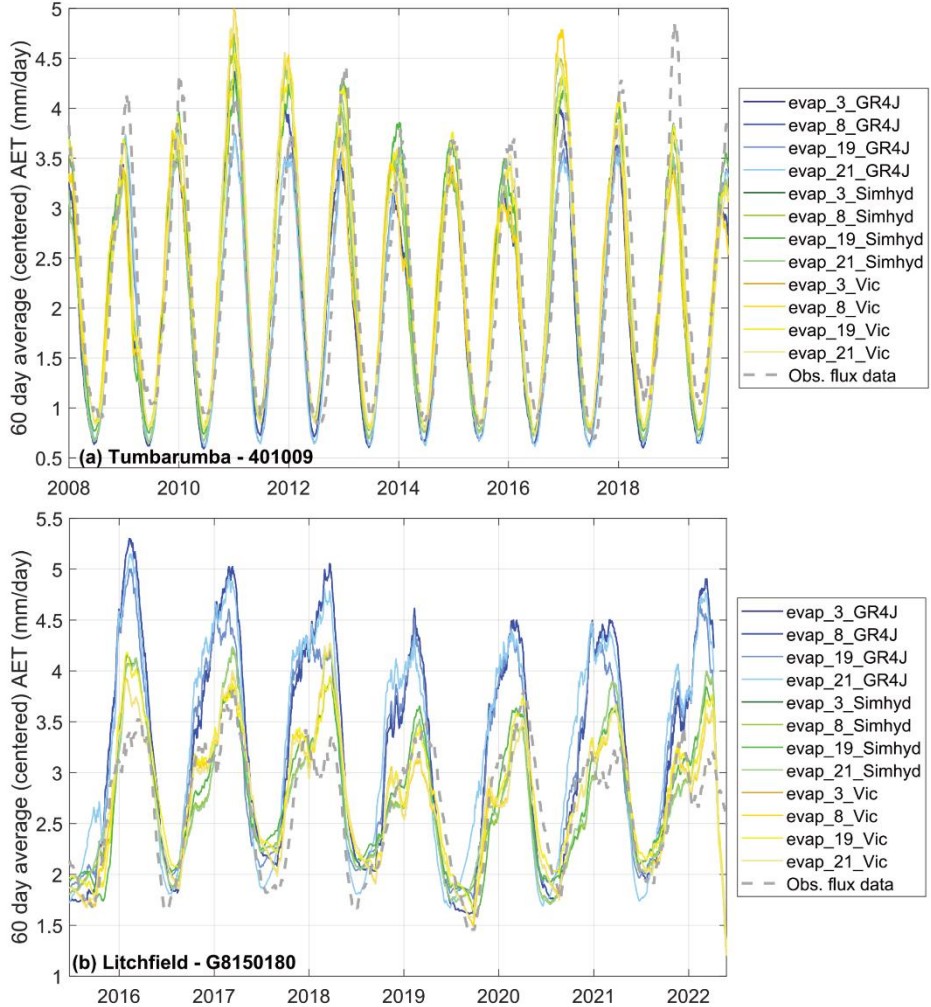

**Figure 5. Modelled AET vs. observed flux tower data, showing the four best performing AET equations across the three models at Tumbarumba (a) and Litchfield (b).**

To explore this discrepancy, GR4J was recalibrated for Litchfield using only AET, giving the model the best possible chance to match observed AET without the constraint of streamflow calibration. However, even under these conditions, GR4J still failed to replicate AET accurately at this site, indicating a possible structural limitation in the model. Results of this analysis are shown in supplementary materials (S3), which includes the performance of all plausible equations under AET-only calibration. No improvement in AET performance was observed compared to the results shown in Fig. 5b.

## 3.2 Signatures

Beyond matching overall OFV scores, AET equations were also assessed using evapotranspiration signatures (Gardiya Weligamage et al., 2025). Similar to hydrological signatures, these metrics break down AET behaviour into distinct





components, focusing on characteristics such as the mean, variability and periodicity of AET.  Across all eight signature plots,
        we observe broadly consistent patterns, so only one representative scatterplot is shown here.

        Below we focus on a timing-based signature, namely monthly AET asynchronicity with PET (Fig. 6), while the remaining
        signatures are shown in the supplementary materials (S4). This signature captures the degree to which AET follows the
        seasonal cycle of PET, independent of magnitude, with greater asynchronicity suggesting a decoupling between the two.

Including this signature helps evaluate whether equations reproduce not just how much evapotranspiration occurs, but when it
        occurs—a key aspect of improving process representation in conceptual models. For example, high asynchronicity might
        suggest that a model fails to capture stomatal regulation or delayed transpiration responses to atmospheric demand, processes
        that are critical under drought or seasonal stress.

        The results indicate that while some equations clearly perform poorly, there are a few that consistently align well with the

observed signatures, including across multiple metrics. These better-performing equations also generally agree with those
        identified through OFV-based testing. Notably, Equation 19 is among the top performers across all signature comparisons.

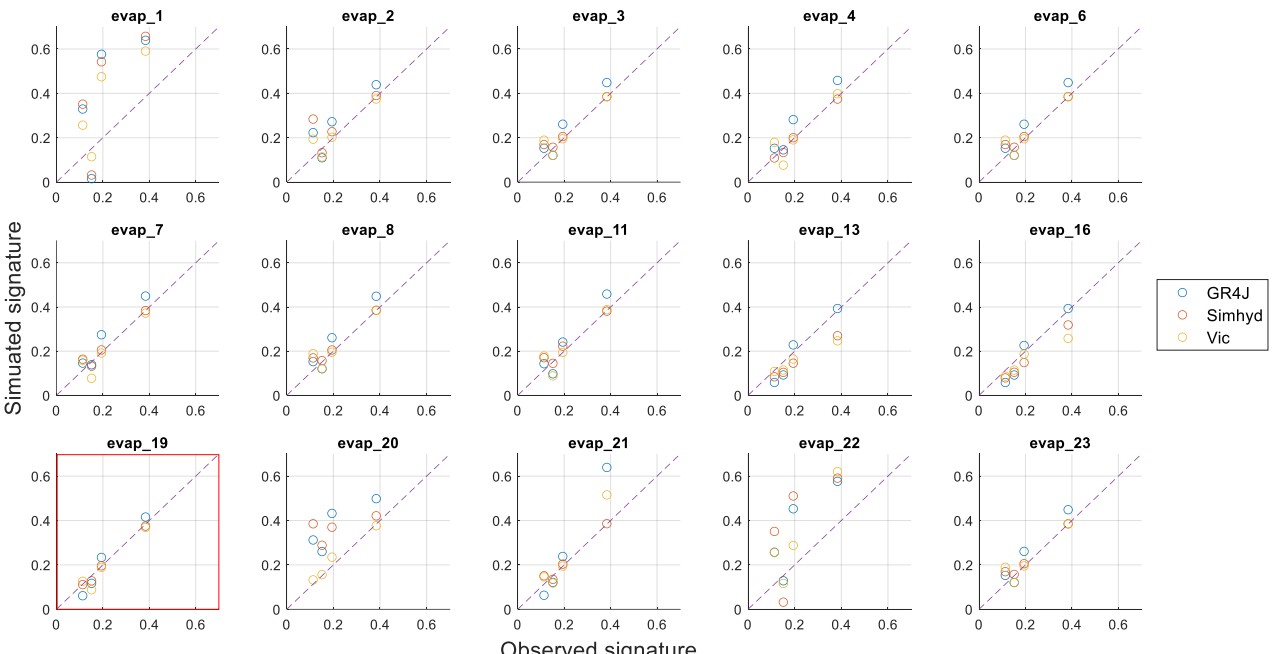

**Figure 6. Monthly asynchronicity of AET: observed vs. simulated, across all equations and models. Equation 19 is highlighted with
a red box. Remaining signature plots are in supplementary materials S4. This signature is purely based on the asynchronicity
between normalised PET and AET, calculated by quantifying the area between the normalised curves, and is indicative of seasonal
water stress.**






### 3.3 Split sample test of equation 19

As an evaluation of robustness, a split-sample test was conducted, comparing the original model formulations with those incorporating the best-performing evapotranspiration equation, AET equation 19 (Fig. 7). Each model was calibrated on one
half of the data and evaluated on the other, with results reported separately for AET and streamflow (Q).

For the calibration period, (and as shown previously), models using evap_19 achieved higher AET objective function values (OFVs) across all catchments compared to the base equations (Fig. 7a). Streamflow performance also generally improved under evap_19 during this period (Fig. 7b). As for the evaluation period for AET, equation 19 consistently improved performance, maintaining higher OFVs across most catchments, similar to the trend observed in calibration (Fig. 7c).

For streamflow, the evaluation period results were more variable. While some catchments showed improvements with evap_19, others performed similarly or worse than the original formulation (Fig. 7d). Notably, catchments that performed poorly under the original AET equation also tended to have low performance under evap_19.

It is important to acknowledge that the available flux tower data covers a relatively short timeframe, which may limit the reliability of this test in fully capturing long-term model behaviour (see supplementary materials S5). This constraint could, in
part, explain discrepancies in streamflow performance during evaluation. Despite this, the results provide evidence that integrating evap_19 enhances AET representation and, in many cases, improves model performance beyond AET.



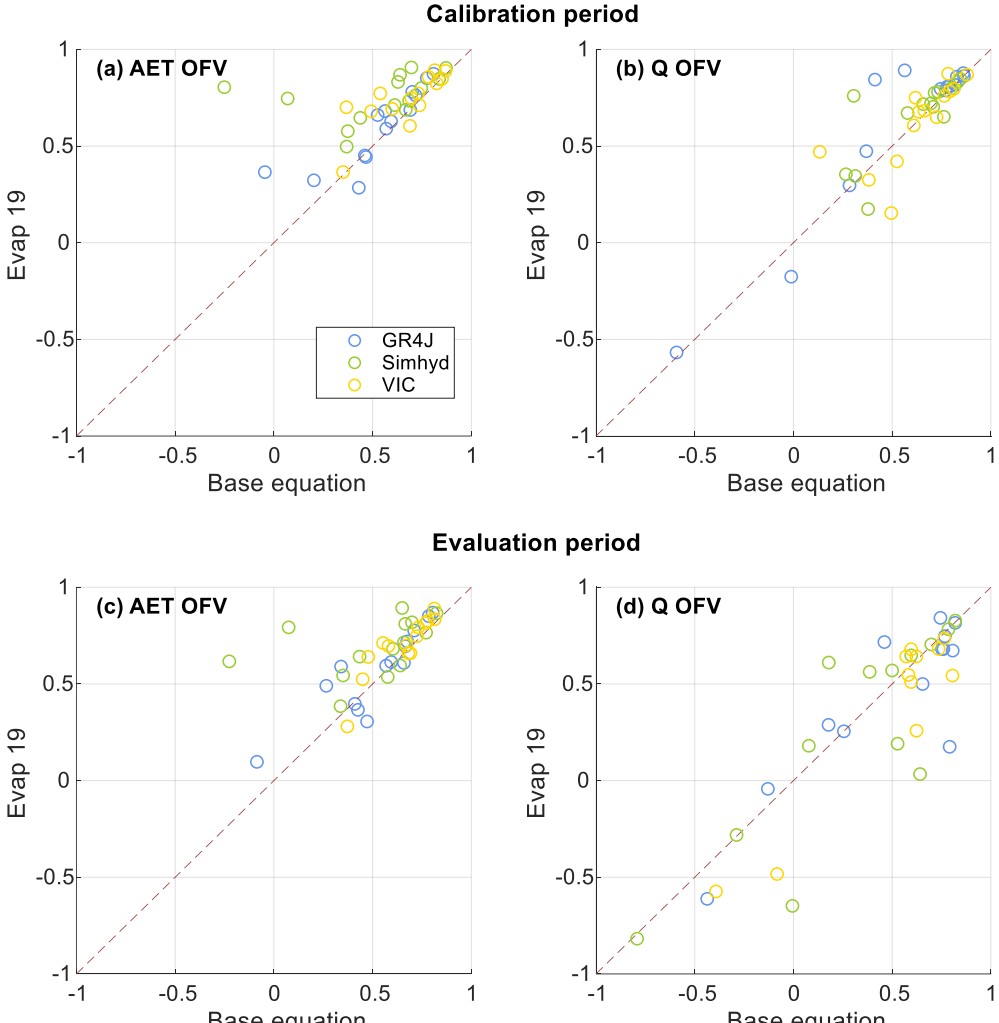

**Figure 7. Split-sample test results showing objective function values for AET and streamflow (Q). Each point represents a model–catchment combination calibrated on one half of the data and evaluated on the other. Panel figures are as follows: results during the calibration period for AET (a) and streamflow (b), and results during the evaluation period for AET (c) and streamflow (d). Note: Streamflow points with OFVs < −1 under the base equation are omitted.**

### 3.4 Seasonal timing of AET

Overall, with an appropriate choice of equation, the models capture AET dynamics relatively well compared to the default options. However, a key feature that emerged upon closer examination of the time series (Fig. 5) was that modelled actual evapotranspiration (AET) appeared to peak earlier than observed flux tower data. This pattern is also apparent in most other catchments (see supplementary materials S1). To investigate this further, we expanded the analysis of the "monthly peak" signature. Specifically, we examined a 7-month window centered on the observed peak month (i.e., three months before, the



peak month itself, and three months after). Based on the time series patterns, it appeared that the models were overestimating AET before and during the peak while underestimating it afterward.

To test this, we quantified the proportion of AET occurring in the three months leading up to and including the peak, expressed as a percentage of the total AET over the 7-month period. This metric was then used to assess how well the models reproduced the observed AET distribution. The calculation was focused on the best-performing evaporation equation (evap_19), for the three hydrological models across the seven catchments under the multi-objective calibration to both streamflow and AET (Fig. 8.a).

To determine whether this timing mismatch was due to model structure, we repeated the analysis calibrating solely to AET to assess whether the models had the capacity to match the observed timing more accurately (Fig. 8.b). Finally, we calibrated to streamflow alone to evaluate how much the AET timing deteriorated when AET was not explicitly considered in the calibration process (Fig. 8.c). The results of this analysis are presented in Fig. 8, which confirms the initial impression about early timing of simulated AET. Further, it shows that although the AET distribution improves when calibrating to AET alone, seasonal

AET timing remains a clear limitation of the models, even when utilising evap_19.




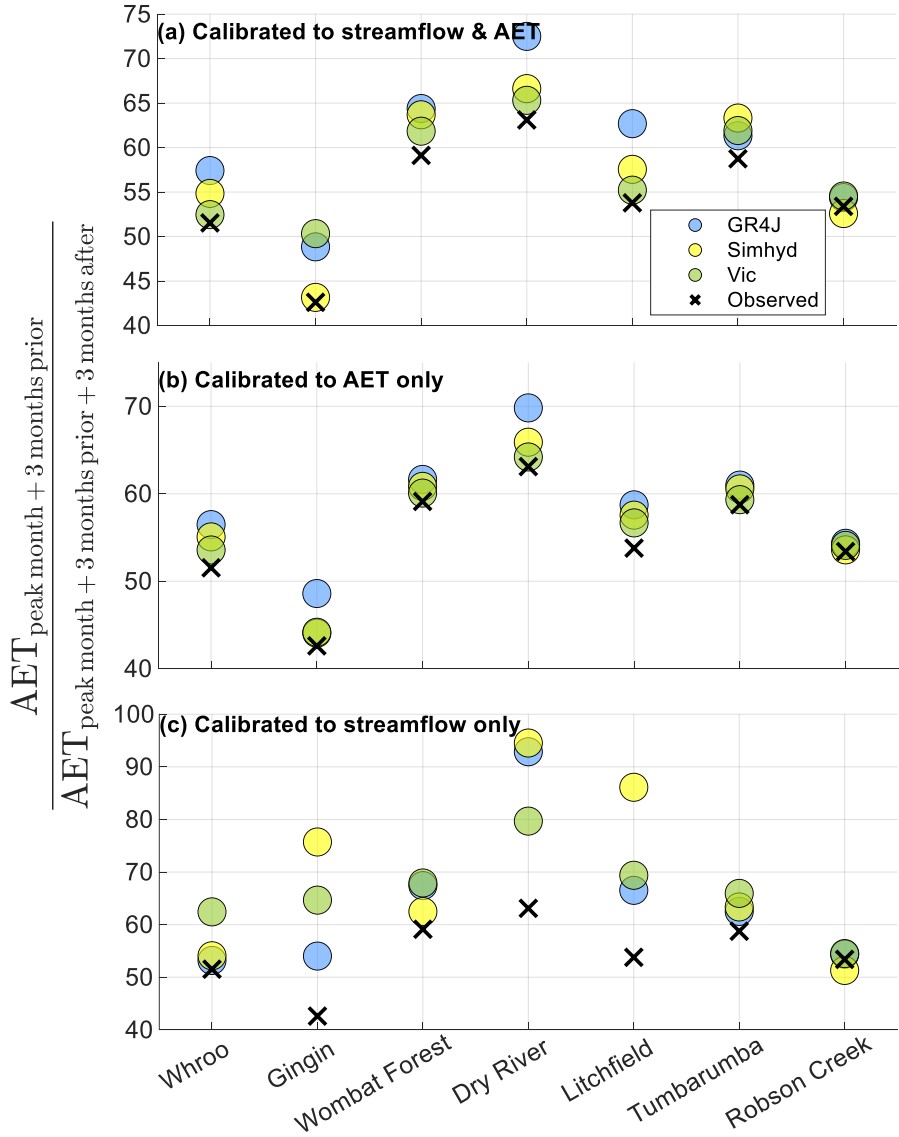

**Figure 8. Percentage of AET which occurs in the first half of the observed seasonal peak, for each catchment and model combination, utilising evapotranspiration equation 19. The models are calibrated to (a) streamflow and AET, (b) AET only, and (c) streamflow only. Note: The 7-month period is determined by identifying the peak month of observed AET (the most common peak month across the flux data time period—one of the AET signatures). The values shown represent the total AET accumulated in the first four months (including the peak month) as a percentage of the total AET over the full 7-month period. Catchments are ordered by increasing total AET over the 7-month period.**




## 4 Discussion

### 4.1 Overview of findings

This study systematically tested 15 AET equations within three widely used conceptual rainfall-runoff models across seven catchments with nearby flux tower AET data. This responds to the call for evaluation of evapotranspiration in hydrological models in previous work (e.g. Kelleher & Shaw, 2018; Dembélé et al., 2020), by isolating equation performance from model structure. Our results show that, while absolute performance varied across catchments and models, a small subset of AET equations, particularly Equation 19, consistently ranked among the top performers relative to the others. Equations 3, 8, and

21 also performed well in some cases but exhibited more variability across models and sites.

While this study primarily focuses on AET equation performance rather than inter-model comparisons, some patterns emerged. For example, GR4J struggled to replicate AET at certain sites (e.g., Litchfield), even when recalibrated using AET-only optimisation. This suggests possible structural limitations in GR4J's partitioning of available water between AET and runoff, highlighting the importance of model-specific considerations when selecting or modifying AET equations.

The main remaining issue across models was that AET often peaked earlier than observed in the flux tower data. While Equation 19 consistently outperformed the others, it only partially addressed this problem—reducing but not eliminating the seasonal timing mismatch. Below, we first examine why equation 19 provides superior simulations, before unpacking the broader implications, both for equation 19 and the remaining challenges highlighted by this study.

### 4.2 Why is equation 19 best?

To understand why Equation 19 provides improved simulations, we first need to examine how it differs from other formulations. The structure of the equation is:

$$AET = \min \left( S, PET, p_1 \times PET \times \left( \frac{S}{Smax} \right)^{p_2} \right), (4)$$

Two key features stand out: (i) its ability to restrict AET to below PET, even under high soil moisture conditions, via the $p_1$ parameter; and (ii) its concave, non-linear relationship between AET and relative soil moisture, shaped by the $p_2$ parameter.

Both parameters are restricted between 0 and 1.

The $p_1$ parameter introduces a physically meaningful limitation, reflecting that vegetation or atmospheric conditions often prevent actual evapotranspiration from reaching its potential rate.

The $p_2$ parameter governs how AET responds to soil moisture: when $p_2 = 1$, the relationship is linear; when $p_2 < 1$, the relationship becomes concave, meaning AET increases rapidly at low soil moisture levels but tapers off as soil moisture

becomes abundant. This behaviour appears to better reflect vegetation function in these catchments, where plants actively transpire when water is limited but taper water use under wet conditions, aligning with the need for more physically grounded representations of vegetation-mediated water use (Deb & Kiem, 2020).





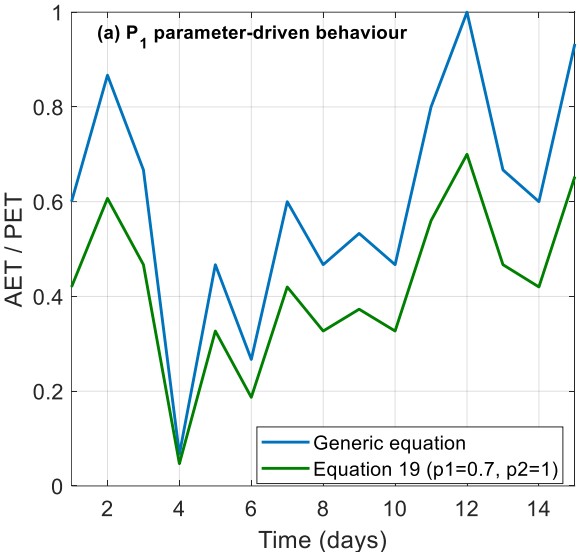
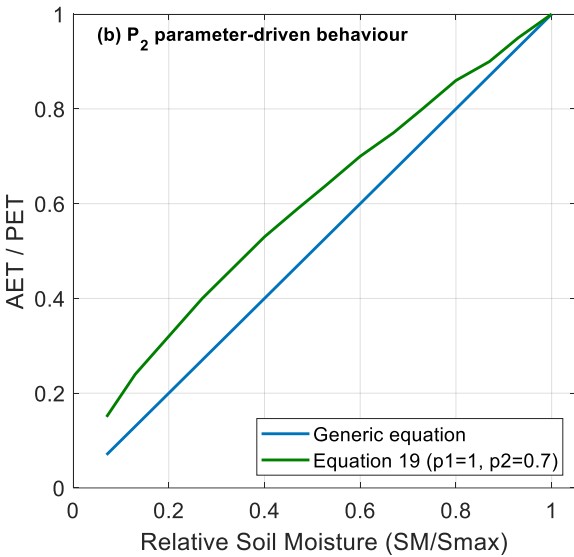

**Figure 9. Comparison of Equation 19 with a typical evapotranspiration formulation (here the generic equation is represented as equation 19 where p1 and p2 equal 1). (a) $p_1$ parameter-driven behaviour: Time series of AET/PET over 15 days, illustrating how Equation 19 constrains AET to remain below PET. (b) $p_2$ parameter-driven behaviour: Relationship between relative soil moisture (SM/S$_{max}$) and AET/PET, showing the stronger non-linear response of Equation 19 to changes in soil moisture availability.**

The calibrated values of these parameters across the seven catchments and three models (see supplementary materials S6) show that $p_1$ values were generally well below 1.0, indicating that this PET-limiting feature was frequently required to match observed AET patterns. For instance, in VIC, $p_1$ ranged from 0.28 (Gingin) to 1.00 (Robson Creek), with similarly constrained values in GR4J and Simhyd. This suggests that the models often needed to invoke sub-PET limitations, which may reflect transport limitations on evapotranspiration, although further evidence is needed to confirm this. The $p_2$ values also varied between catchments, often falling below 0.5 (e.g., Wombat Forest, Tumbarumba), indicating the need for a concave soil moisture response. That is, AET increases quickly under dry to moderate conditions but becomes less sensitive as soil moisture approaches saturation. While some evapotranspiration equations assume AET will continue increasing proportionally with soil moisture, this behaviour appears inconsistent with the flux tower data and the known role of vegetation in regulating water loss.

Together, these features help explain why Equation 19 outperformed alternatives: it avoids the assumption that AET always equals PET under moist conditions, reduces over-extraction of water, and better captures the dynamic relationship between soil moisture and evapotranspiration observed in flux tower data.

Conversely, the poorer-performing AET equations exhibited several recurring issues. Many lacked explicit constraints or responded too linearly to changes in soil moisture, resulting in rapid water depletion and elevated AET values that rarely match flux tower behaviour. Several formulations allowed AET to equal PET frequently—an outcome that overlooks the regulatory role of vegetation and other limiting processes.



However, despite Equation 19's relative improvements, it does not fully resolve all issues. As shown in Fig. 8, it still overestimates AET in early-season periods. This suggests our conceptual models are lacking the knowledge that water availability does not lead to immediate transpiration. Equation 19 mitigates this issue more effectively than the other tested equations, but the challenge remains.

## 4.3 Implications

Accurate AET representation is essential for reliable hydrological modelling, particularly when applied to future climate scenarios (Zhao et al., 2013). Incorrect AET modelling could propagate errors, leading to misleading projections of water availability, catchment response, and long-term water balance estimates. Ensuring that AET equations appropriately constrain AET to realistic values is therefore required to improve model robustness and predictive capacity.

The findings of this study highlight how assumptions embedded within many conceptual models, especially linear or overly
simplified relationships between PET and soil moisture, can lead to systematic biases in AET simulations. These relationships are often taken for granted rather than being critically evaluated, despite their substantial influence on model behaviour. By drawing attention to this issue, we hope to encourage deeper scrutiny of the ways AET is represented within hydrological models, particularly in light of observed vegetation responses (Duethmann et al., 2020).

The strong performance of Equation 19 could potentially be attributed to its origin of purpose, in the conceptual hydrological
model LASCAM, which, during its creation, included thought on the inclusion of vegetation impacts (Sivapalan et al., 1996). This included the interaction of the deep-rooted eucalyptus trees found within Australia, which led to the use of constraints on soil moisture availability and AET behaviour, which (as demonstrated here) are more realistic when evaluated against flux tower data. These insights suggest that existing models could benefit from replacing simplistic AET equations with more process-informed alternatives or by developing new formulations inspired by Equation 19.

While Equation 19 consistently improved model performance, its limitations offer insight into deeper ecohydrological processes that remain unaccounted for, such as the seasonal partitioning of AET. Although Equation 19 better matches overall AET signatures, it does not fully capture the observed differences in AET between the beginning and end of the season. Specifically, the catchments appear to be photosynthesising at a lower-than-expected rate (using less water and thus exhibiting lower AET) early in the season. This discrepancy likely arises because conceptual models, which lack explicit vegetation
components, assume that water availability directly translates into immediate AET increases. In contrast, observed data suggest that the catchment does not use all available water at the start of the season, possibly due to physiological constraints on vegetation growth. If plants transpired at the rate the model predicts, they could experience excessive growth that would not be sustainable through the dry season. Similar findings have been reported in previous studies, such as Stephens et al. (2025 – in prep), which found that catchments in intermediately wet regions (i.e. not arid or very wet) often exhibited lower-than-
expected AET in wet periods. Lower-than-expected wet season transpiration was also demonstrated by Eamus et al. (2000) in savannah trees, suggesting that  the vegetation's capacity to transport moisture may limit AET during the wet season.



The problems we found with seasonal timing of AET indicate that, even with improved AET parameterisation, conceptual models may still lack a critical physical mechanism governing seasonal water use. Future work should explore ways to
incorporate such ecohydrological feedbacks into hydrological models, ensuring that the representation of AET accounts for vegetation constraints and long-term water availability strategies. Additionally, further studies should investigate whether this catchment behaviour is unique to Australian ecosystems or occurs more broadly. Australia's high interannual climate variability may encourage conservative water use strategies in vegetation that differ from those in more temperate or consistently wet climates (Norton et al., 2022). Exploring such geographic differences could help tailor model structures to
better reflect regional vegetation–climate interactions.

Moving forward, systematic evaluations of internal flux equations in hydrological models should become standard practice, particularly when combined with multi-objective calibration approaches that leverage real-world data such as flux tower observations. Although we recognise limitations associated with flux tower coverage, using such data helps highlight where improvements to model structure, rather than parameterisation alone, may be needed. Additionally, further integration of
empirically driven, process-based enhancements in hydrological models could help refine AET representation. These improvements would aid hydrologists in selecting AET equations that best match their specific modelling objectives, ultimately enhancing the reliability of hydrological simulations across diverse environmental conditions.

Our study's inclusion of many models, catchments, and AET equations provides a comprehensive assessment, but future work should explore individual model performance in greater detail. A logical next step would be to test Equation 19 within all 47
available hydrological models, using the same methodology applied here, to assess whether similar performance trends hold across a greater range of models. Additionally, this study did not evaluate AET equations in deficit-style hydrological models. Future research should extend this analysis to these alternative modelling frameworks to determine whether similar AET performance patterns emerge, particularly given evidence of more realistic AET dynamics in deficit models in water limited conditions (Fowler et al., 2021).

Lastly, we note the limitation discussed above, namely that the best-performing AET formulation—which we recommend other researchers test and, if appropriate, adopt—does not entirely solve the issues regarding seasonal timing of AET in the catchments tested here. Continued exploration of equation-level performance in controlled testing frameworks, like the one used here, will help to identify process representations that bridge conceptual and process-based modelling—a critical direction for future model development (Knoben et al., 2019; McMillan, 2021).

**Conclusion**

This study conducted a systematic evaluation of 15 actual evapotranspiration (AET) equations within three conceptual hydrological models across diverse Australian catchments. By isolating AET formulations while holding other model components constant, this study was better able to identify variations in performance due to the AET equations themselves (as distinct from the surrounding model structure). Equation 19 consistently outperformed alternatives in both streamflow and

AET objectives, as well as AET signature alignment. Its success appears linked to its non-linear soil moisture dependence and explicit limitation of AET below potential evapotranspiration, aligning better with observed flux tower data.

Despite this improvement, persistent mismatches in seasonal AET timing, especially early-season overestimation, highlight limitations in current conceptual models. These results suggest that empirical equations alone may be insufficient to fully capture vegetation-driven dynamics, especially under conditions of climatic or phenological change. While substituting AET

equations such as Equation 19 can enhance performance, future work should aim to integrate ecohydrological mechanisms, such as plant water regulation and delayed transpiration responses, into model structures.

Overall, this research demonstrates the importance of critically assessing and selecting AET equations in conceptual modelling. Incorporating process-informed empirical equations and advancing the representation of vegetation dynamics will be essential for improving the robustness of hydrological simulations by ensuring accurate water partitioning, particularly under changing

environmental conditions.

**Author contribution**

GB: Conceptualisation, data curation, formal analysis, investigation, methodology, validation, visualisation, writing – original draft, review and editing.

KF: Conceptualisation, methodology, supervision, writing – review and editing

MP: Conceptualisation, methodology, supervision, writing – review and edit

CS: Conceptualisation, methodology, supervision, writing – review and edit

**Competing interests**

The authors declare that they have no conflict of interest.

**Acknowledgements**

This research was supported by an Australian Government Research Training Program Scholarship and an Ingenium Scholarship provided by the Faculty of Engineering Information and Technology, The University of Melbourne.

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
