# Peer review of "A systematic evaluation of 15 actual evapotranspiration formulations within conceptual hydrological models - supplementary materials"

_EGUsphere, 2025_

## Author Comment (AC1)

Reviewer comments, response and actions for manuscript:

**A systematic evaluation of 15 actual evapotranspiration formulations within conceptual hydrological models**

Co-authors: Gabrielle Burns, Keirnan Fowler, Murray Peel, Clare Stephens

**Reviewer 1:**

*"Burns et al. present a study on evaluating 15 different actual evapotranspiration methods within three conceptual hydrological models across seven diverse Australian basins. They establish a multi-objective calibration framework, in which observed streamflow and observed AET data from flux tower sites are used. The authors have established a comprehensive framework, and their current manuscript might be suitable for publication in HESS after addressing my comments listed below. The readability of some figures could be enhanced by increasing the font size of the labels. Tables are not well-designed, and it is often hard to identify the information to which row the appropriate information belongs. Please, revise. Overall, I value that the authors have considered more than one hydrological model structure and used a joint multi-objective calibration of the model's parameters. Below you will find my three main remarks and further below, more minor suggestions for revisions:"*

| Reviewer 1 | Response | Action |
|---|---|---|
| **Major comments** | | |
| I missed any discussion on the spatial transferability of the calibrated parameters. I understand that this methodology is presented as site-specific. IN the line of Klemes et al., 1986 (doi 10.1080/02626668609491024), to assess robustness of the most suitable AET formula, I would also definitely welcome an evaluation of the selected parameters in ungauges locations (apply the 1calibrated parameter set to the 6 remaining basins) and objectively assess, whether the selected AET formula also holds across other locations as well. This could enhance the temporal split-sample test, already presented. | We thank the reviewer for raising the important issue of parameter transferability and for referring to the validation framework of Klemeš (1986). Following the logic of Klemeš, the proposed proxy-basin test is valuable when the objective is prediction in ungauged basins using a fixed model structure and transferable catchment-scale parameters.

We would first like to clarify that the split-sample testing presented in this study refers to temporal split-sample validation, rather than a spatial or proxy-basin test. We will revise the manuscript to explicitly state this (i.e. temporal split-sample testing) to avoid any ambiguity and to make clear that spatial transferability is not assessed here.

While we agree that a proxy-basin experiment can provide useful insights, the spatial transferability of conceptual models is not our focus. As per our existing abstract, our focus is temporal transferability ("models [may] incorrectly simulate | We consider spatial transferability and proxy-basin testing to be a valuable but distinct line of inquiry, particularly relevant for studies explicitly targeting regionalisation or prediction in ungauged basins. We will acknowledge this more clearly in the discussion as an important avenue for future work, but that it lies beyond the intended scope of the present manuscript. |

| Reviewer 1 | Response | Action |
|---|---|---|
| | long-term changes in partitioning between AET and streamflow"). Spatial and temporal transferability are distinct challenges due to the known site-dependence of evaporative and vegetative processes. This means that poor spatial transferability of calibrated parameters would not necessarily indicate a lack of robustness of a given AET formulation but rather reflect differences in vegetation across space. Thus, there is little value in testing this transferability without an explicit framework to handle these spatial differences. Such a framework would bloat the paper, which is already long (30 pages, 9 figures) and is quite complex already (3x model structures, 15x equations and 7 catchments). | |
| When focusing on AET, the current formulation does not consider any uncertainty in potential evapotranspiration estimates. The authors should consider their in the discussion, see e.g. studies like Oudin et al, 2005 (JoH), Pimentel et al, 2023 (WRR); Thakur et al, 2025 (HESS); which presents sometime large differences of employed PET methods on hydrological components from different perspectives, besides uncertainty in the precipitation estimates, which is also a crucial element for the water balance closure. | We agree that PET formulation and precipitation uncertainty are important contributors to uncertainty in hydrological simulations. In this study, the decision to adopt a single PET formulation was intentional, as our aim was to isolate the effects of alternative AET formulations rather than to explore the full range of forcing uncertainty. Nevertheless, we agree that the implications of PET formulation uncertainty should be acknowledged more explicitly in the manuscript, and we will expand the discussion accordingly.

In addition, to directly address this comment, we have included a targeted sensitivity analysis examining the effect of an alternative PET formulation on one of the key diagnostic results. Specifically, we reproduced Figure 8 using an alternative PET forcing (Morton's Point Potential), with all three models re-calibrated across the seven catchments. This additional figure will be provided in the Supplementary Materials and referenced in the revised discussion. | Add in to discussion PET uncertainties & omission of testing different PETs. Example:

"An important source of uncertainty not explicitly explored in this study is the estimation of potential evapotranspiration (PET). Numerous studies have demonstrated that different PET formulations can produce different estimates of atmospheric evaporative demand, with implications for simulated evapotranspiration, soil moisture, and runoff (e.g. Oudin et al., 2005; Pimentel et al., 2023; Thakur et al., 2025). In the present analysis, |

| Reviewer 1 | Response | Action |
|---|---|---|
| | | PET formulation was held constant across all experiments in order to isolate the effects of AET formulation specifically.

One additional analysis in reference to this issue was investigated by not reproducing all the results for a different PET formulation, but rather we reproduced Figure 8, which highlights the systematic tendency for simulated AET to peak earlier in the season than observed, using an alternative PET input (Morton's Point Potential). For this experiment, all three models were re-calibrated across the seven catchments using the alternative PET forcing.

While some catchment-specific differences in the magnitude of the response were observed (with slightly stronger or weaker signals), the overall patterns and conclusions remained unchanged. The timing bias in simulated AET persisted across PET formulations, indicating that the early seasonal peak identified in the original analysis is not a consequence |

| Reviewer 1 | Response | Action |
|---|---|---|
| | | of the choice of PET input. Given the overall outcomes were relatively robust despite the very different PET formulation, we feel this is a sufficient response to this valid concern. |
| It is not very clear, at this temporal resolution the study was done. Daily? I missed information about the length of the calibration periods in the Methods. This should be provided around Lines 241-247, to provide more details on the split-sample testing. | We thank the reviewer for noting this lack of clarity. The simulations were performed at a daily time step, which was not stated explicitly in the Methods and will be added for clarity.

We also agree that additional detail on the calibration and split-sample periods should be provided in the Methods section around Lines 241–247. The length of the calibration periods differs among sites, reflecting the varying availability of flux tower observations. While this information is currently described in the Supplementary Material, we will add a brief description in the Methods and clearly reference the Supplementary Table where site-specific calibration periods are reported.

As noted in the Results (Lines 328–329), the relatively short duration of flux tower records at some sites may limit the ability of the split-sample test to fully capture long-term model behaviour. We will ensure this limitation is clearly introduced earlier in the Methods when describing the split-sample testing procedure. | Add detail about flux tower time lengths for split sample test around lines 241-247.

Also add in that it was a daily time step in the introduction. |
| Minor comments | | |
| Lines 28-29: be careful, soil moisture is not internal model's flux, please, reformulate sentence accordingly, soil moisture is a state! | We thank the reviewer for this clarification and agree. The sentence will be revised to distinguish between simulated state variables (e.g. soil moisture) and fluxes (e.g. groundwater recharge, AET). | Revise wording accordingly. |

| Reviewer 1 | Response | Action |
|---|---|---|
| First paragraph of Introduction: please, say more clearly, which kind of hydrological models you have in mind (conceptual, black box, spatially distributed, etc, there are many subcategories ). | We agree that additional clarity is helpful. While this study focuses on conceptual rainfall–runoff models, the statements in the opening paragraph are intended to apply more broadly to hydrological models in general. We will revise the text to make this distinction explicit. | Clarify model classes in Introduction. |
| Equation (4) should be written in full in the Methods, providing a reference to original work is not enough here. | Yes agreed, the full function is:

E = E_of - 5 * abs(ln(1 + B)) ^ 2.5 where B = mean(sim - obs) / mean(obs)

We agree and will include the full formulation of the objective function in the Methods. | Add full equation and definitions. |
| Line 30: remove "perhaps" and rephrase. | We agree and will rephrase the sentence to avoid speculative language while retaining the intended meaning. | Reword sentence. |
| Line 31: instead of "culmination", consider using "integral variable" | We appreciate the suggestion and will revise the wording to better reflect that streamflow integrates multiple catchment processes. | Revise wording. |
| Consider merging the last two paragraphs of Intro, and move the detailed MARRMOT descriptions to the Methods, where it fits better. | We agree that this would improve structure and readability and will revise accordingly. We will keep the aim in the introductory section, and move the remaining material in these paragraphs to the methods, as suggested. | Reorganise Introduction and Methods. |
| Line 140: you don't say, which methods was used to calculate PET. This is important. Also, in section 2.3, information on the discharge is missing. Additionally, here is the right place to introduce length of the timeseries, number of years, and say, how did you treat missing values in your analysis. | PET is calculated using Morton's wet-environment formulation as provided in the CAMELS dataset; we will make this explicit. Streamflow data are also sourced from CAMELS and will be clearly stated. We will add details on the temporal coverage of the datasets and note that missing values were excluded from the analysis. | Include sentence explicitly stating that streamflow data is from CAMELS and PET is Morton's Wet environment.

Add details that say the CAMELS dataset is from 01-Jan-1950 to 26-May-2022, and that missing values were ignored. The dates of the flux tower timeseries can be seen in the supplementary materials table S5. |

| Reviewer 1 | Response | Action |
|---|---|---|
| Line 268: equation 19 should be connected to the AET definition, Current statements looks like it refers to eq. 19, which is not defined in the manuscript. | Noted. References to AET equations will be updated to use consistent naming aligned with the tables (e.g. evap_19). | Update equation references throughout |
| Table 4: instead of having only categorical values for which AET method was best, it would be also valuable to have some qualitative category. | We note this suggestion. Quantitative performance metrics are already provided in the Supplementary Material. Can the reviewer please provide feedback as to whether this is what is meant? | We will add text to make it clearer that this quantitative information is in the Supplementary Material. This will be added in the caption of Table 4 and associated text. |
| Figure 4: Consider different colors for native equation: now, every model's native equation is shown by black star, with overlaying colored circle. Considering coloring the star by model's color. Also, I recommend not to use 4 columns, rather 3 columns and then 3 rows. This would allow your figures to become bigger | We agree and will revise the figure accordingly, including exploring an alternative subplot layout to improve readability. | Update figure to have the stars as the model colours (perhaps outlined in black to distinguish?)

Adjust composition of subplots to better show as per recommendation. |
| Figure 6: I would recommend to keep same model's colors as used earlier in Figure 4. | We thank the reviewer for highlighting this inconsistency. In Figure 6, the Simhyd model should indeed be shown in green, consistent with the colour scheme used elsewhere in the manuscript (e.g. Figure 4). We have corrected this in the revised figure and updated the caption accordingly. | Update Figure 6 to ensure consistent colouring with the rest of the manuscript. |

**Additional Figure:**

Comparison of Figure 8 under alternative PET formulations.

The left panel reproduces the original Figure 8, showing the systematic overestimation of AET during the first four months of the seven-month peak window, with models calibrated using Morton's Wet Environment PET.

The right panel shows the same analysis with all models re-calibrated using Morton's Point Potential PET. The left-panel figure will be included in the Supplementary Materials of the revised manuscript.